# Menstrual Blood-Derived Stem Cell Paracrine Factors Possess Stimulatory Effects on Chondrogenesis In Vitro and Diminish the Degradation of Articular Cartilage during Osteoarthritis

**DOI:** 10.3390/bioengineering10091001

**Published:** 2023-08-24

**Authors:** Ilona Uzieliene, Paulina Bialaglovyte, Rokas Miksiunas, Ignas Lebedis, Jolita Pachaleva, Raminta Vaiciuleviciute, Almira Ramanaviciene, Giedrius Kvederas, Eiva Bernotiene

**Affiliations:** 1Department of Regenerative Medicine, State Research Institute Centre for Innovative Medicine, LT-08406 Vilnius, Lithuania; paulinabialaglovyte@gmail.com (P.B.); rokas.miksiunas@imcentras.lt (R.M.); ignaslebedis@gmail.com (I.L.); jolita.pachaleva@imcentras.lt (J.P.); raminta.vaiciuleviciute@imcentras.lt (R.V.); eiva.bernotiene@imcentras.lt (E.B.); 2Department of Immunology, State Research Institute Centre for Innovative Medicine, LT-08406 Vilnius, Lithuania; almira.ramanaviciene@imcentras.lt; 3NanoTechnas—Center on Nanotechnology and Materials Sciences, Faculty of Chemistry and Geosciences, Vilnius University, LT-03225 Vilnius, Lithuania; 4The Clinic of Rheumatology, Traumatology Orthopaedics and Reconstructive Surgery, Institute of Clinical Medicine of the Faculty of Medicine, Vilnius University, 03101 Vilnius, Lithuania; giedrius.kvederas@santa.lt; 5Department of Chemistry and Bioengineering, Faculty of Fundamental Sciences, VilniusTech, Vilnius Gediminas Technical University, 10223 Vilnius, Lithuania

**Keywords:** menstrual blood mesenchymal stem cells, bone marrow mesenchymal stem cells, chondrogenic differentiation, TGF-β3, activin A, BMP-2, IGF-1, cartilage, paracrine effects

## Abstract

Articular cartilage is an avascular tissue with a limited capacity for self-regeneration, leading the tissue to osteoarthritis (OA). Mesenchymal stem cells (MSCs) are promising for cartilage tissue engineering, as they are capable of differentiating into chondrocyte-like cells and secreting a number of active molecules that are important for cartilage extracellular matrix (ECM) synthesis. The aim of this study was to evaluate the potential of easily accessible menstrual blood-derived MSC (MenSC) paracrine factors in stimulating bone marrow MSC (BMMSCs) chondrogenic differentiation and to investigate their role in protecting cartilage from degradation in vitro. MenSCs and BMMSCs chondrogenic differentiation was induced using four different growth factors: TGF-β3, activin A, BMP-2, and IGF-1. The chondrogenic differentiation of BMMSCs was stimulated in co-cultures with MenSCs and cartilage explants co-cultured with MenSCs for 21 days. The chondrogenic capacity of BMMSCs was analyzed by the secretion of four growth factors and cartilage oligomeric matrix protein, as well as the release and synthesis of cartilage ECM proteins, and chondrogenic gene expression in cartilage explants. Our results suggest that MenSCs stimulate chondrogenic response in BMMSCs by secreting activin A and TGF-β3 and may have protective effects on cartilage tissue ECM by decreasing the release of GAGs, most likely through the modulation of activin A related molecular pathway. In conclusion, paracrine factors secreted by MenSCs may turn out to be a promising therapeutical approach for cartilage tissue protection and repair.

## 1. Introduction

The loss of articular cartilage extracellular matrix (ECM) is a hallmark of osteoarthritis (OA) and follows catabolic processes and degradation of the tissue. The pathogenesis of OA involves mechanical, metabolic and inflammatory factors, which ultimately leads to structural joint destruction [1,2]. Currently, no effective therapy, as well as disease modifying drugs for OA and cartilage tissue repair, exist. Mesenchymal stem cells (MSCs) seem promising for cartilage tissue regeneration due to their potential to differentiate into chondrogenic lineage [3]. For instance, bone marrow MSCs (BMMSCs) are a classical stem cell type which has been mostly studied and used for cartilage tissue repair after trauma or OA [4,5]. The initial step of in vivo chondrogenesis starts from stem cell condensation and differentiation initiation by growth factors. The most important growth factors for chondrogenic differentiation are transforming growth factor βs (TGF-βs), bone morphogenetic matrix proteins (BMPs), insulin growth factors (IGFs), activins, etc. [6,7,8,9]. Therefore, to stimulate MSCs differentiation in vitro, a balance according to growth factors should be assessed. The typical protocol for inducing chondrogenesis in BMMSCs includes TGF-β3. However, the major issues regarding BMMSCs use in tissue engineering are their invasive isolation and small amounts of cells in extracted bone marrow [10]. Therefore, other sources of MSCs are being considered as cell therapies for cartilage. 

Menstrual blood MSCs (MenSCs) are easy to collect without any invasive interventions. MenSCs are much less characterized than BMMSCs, however, they are known to possess more pronounced stem cell properties compared to classical BMMSCs [11,12,13,14]. Also, MenSCs secrete a number of cytokines and growth factors, such as angiogenic factors VEGF, HGF, ANG, and MMP-1, and pro-inflammatory cytokines IL-6, IL-8, and IFN-gamma, and most importantly, MenSCs were shown to be safe to transplant due to their low tumorgenicity [13]. In addition to that, MenSCs do not express the immune activation marker MHC class II antigen (HLA-DR), so they do not cause an immunological reaction and are suitable for allogenic application [15,16,17]. Paracrine effects of MenSCs were analysed in numerous studies and their effects were proposed as being more superior than BMMSCs [14]. For instance, MenSCs paracrine factors showed promising results in rat model of myocardial infarction by reducing the apoptosis of cells and the stimulation of endogenous regeneration [18]. The serum-free condition medium of MenSCs contains a large number of angiogenic factors, such as angiogenin, bFGF, VEGF, HGF, and EGF, which promotes wound healing [19]. In addition to that, MenSCs extracellular vesicle’s (EVs) were isolated and characterized and it was reported they are sufficient to repair rat intrauterine adhesion for embryo. EV effectively recovered the morphology and promoted the regeneration and angiogenesis of the rat glands and stimulated the phosphorylation of SMAD1/5/8 and ERK1/2 [20]. Furthermore, MenSCs EVs were shown to resolve inflammation via induced M1-M2 macrophage polarization and enhanced neoangiogenesis on the wound-healing process in mice [21,22]. MenSCs EVs are currently considered as an effective non-cellular therapy for various tissues. Although MenSCs are an attractive alternative to BMMSCs, their use for cartilage tissue regeneration is still under consideration and research.

The aim of the study was to evaluate the potential of MenSC paracrine factors in stimulating BMMSCs chondrogenic differentiation and their role in protecting cartilage from degradation under inflammatory conditions in vitro.

## 2. Materials and Methods

### 2.1. MenSCs and BMMSCs Isolation and Culture

MenSCs and BMMSCs were isolated and cultured, as previously described [23]. Briefly, menstrual blood samples were collected from 25–35 years old female donors (*n* = 4) using sterile silicone cups (iCare) during the second day of menstruation and MenSCs were isolated using ficol paque (Sigma Aldrich, Merck KGaA, Darmstadt, Germany.) gradient centrifugation. Bone marrow samples were collected after surgical procedures in Santaros Hospital (Vilnius, Lithuania). BMMSCs were extracted from the 28–32 years old female bone marrows (*n* = 3), using a sterile scalpel, then centrifuged and filtered into a sterile tube.

All procedures with the donor tissues were performed in accordance with the Bioethical Permission (No. 158200-14-741) and its supplemented version (Permission No. 158200-741-PP2-34) approved by the Vilnius Regional Biomedical Research Ethics Committee. 

MenSCs and BMMSCs were cultivated under the same conditions in DMEM (1 g/L glucose), supplemented with 10% fetal bovine serum (FBS) (Merck) and 1% penicillin 10,000 units/mL—streptomycin 10,000 μg/mL (PS) (Sigma Aldrich) (named as complete growth medium), 37 °C incubator with 5% CO_2_. For the cell expansion before the experiments, 1 ng/mL of fibroblast growth factor-2 (FGF2) (Thermo Fischer Scientific, Waltham, MA, USA) was added to the MenSCs and BMMSCs medium to maintain their stem cells potential and to avoid spontaneous differentiation. The medium for both cell types was changed twice a week, and the volume of the complete medium was added according to the culture dish used. Cells were re-seeded or were used for experiments by reaching 80% confluence. Early passages (*p* = 2 or 3) of MenSCs and BMMSCs were used in all experiments.

### 2.2. Chondrogenic Differentiation in Pellets with Different Growth Factors

Both MenSCs and BMMSCs were detached, counted, and washed with DMEM medium without FBS by centrifugation at 500× *g* for 5 min. After the cells were transferred into the round bottom, low surface attachment 96 well plates (250,000 cells/well), centrifuged at 500× *g* for 5 min, and the medium was changed to chondrogenic medium, containing high glucose (4.5 g/L) DMEM, 1% PS (Merck), 1% insulin-transferrin-selenium (ITS) (Gibco Life Technologies, Waltham, MA, USA), 0.35 mM L-proline (Carl Roth, Karlsruhe, Germany), 10^−7^ M dexamethasone, 0.17 mM ascorbic acid phosphate (Sigma Aldrich) with or without growth factors. Growth factors—10 ng/mL TGF-β3 (Thermo Fisher Scientific), 50 ng/mL activin A, and 50 ng/mL BMP-2 and 100 ng/mL IGF-1—were added separately and as combinations of TGF-β3 + activin A, TGF-β3 + BMP-2, and TGF-β3 + IGF-1, using the same concentrations, where activin A, BMP-2, and IGF-1 were added only once at the beginning of differentiation. Chondrogenic differentiation was stimulated for 21 days and the chondrogenic medium was changed three times a week. During this time, cells formed pellets. After chondrogenic differentiation, the cell pellets were analyzed by RT-qPCR.

### 2.3. Cell Pellet Gene Expression Analysis

After chondrogenic differentiation, cell pellets were washed twice with PBS and homogenized with a syringe in RLT lysis buffer with 10% mercaptoethanol (from Qiagen RNeasy Mini Kit kit, Venlo, The Netherlands) according to the manufacturer’s recommendations. The RNA was subsequently purified with RNeasy Mini Spin columns (Qiagen kit). RNA concentration and purity were measured with the spectrophotometer (SpectraMax i3 (Molecular Devices, San Jose, CA, USA)). 

RNA samples were treated with dsDNase and cDNA synthesis was performed with the Maxima^®^First Strand cDNA Synthesis Kit (Thermofisher Scientific) according to the manufacturer protocol. RT-qPCR was performed using Maxima Probe qPCR Master Mix (2X) with QuantStudio 1 Real-Time PCR System. The TaqMan Gene Expression Assays were used for gene expression analysis. The RT-qPCR reaction volume contained 25 μL of reaction buffer with 1 μL of 20X Taqman Gene Expression Assay mix. All reactions were run in triplicates. Cycle conditions were as follows: initial denaturation step at 95 °C for 10 min, followed by 40 cycles of denaturation at 95 °C for 15 s and finally annealing and extension at 60 °C for 60 s. Each RNA sample was controlled for genomic DNA contamination by the reactions without reverse transcriptase, and the reagent contamination was checked by the reactions without a template (NTC). For the normalization of gene expression, the geometric mean of two reference genes, RPS9 and B2M, was used. 

### 2.4. Chondrogenic Differentiation in Co-Culture Conditions

The MenSCs were detached, counted, and seeded into 24 well plates (20,000 cells/well) with complete growth medium, while BMMSCs were detached, counted, and seeded into 24 well inserts (0.4 μm pore) (20,000 cells/insert) with complete growth medium. The next day, after cells were attached to the wells/inserts, BMMSCs inserts were transferred into MenSCs seeded wells and chondrogenic differentiation was stimulated with chondrogenic medium with/without 10 ng/mL of TGF-β3. The cells were divided into three groups: MenSCs in wells only, BMMSCs in inserts only, and co-cultures of MenSCs with BMMSCs. All three groups were made in triplicates. Chondrogenic differentiation was stimulated for 21 days and the chondrogenic medium was changed three times a week, collecting the medium for growth factor secretion analysis (ELISA) after 3, 7, and 21 days.

After 21 days of chondrogenic differentiation, the cell pellets were analyzed by cartilage oligomeric matrix protein (COMP) secretion (ELISA). The data were normalized between the groups according to the cell number.

### 2.5. ELISA for Cartilage Oligomeric Matrix Protein

After 21 days of MenSCs, BMMSCs, and their co-culture chondrogenic differentiation, the medium was collected (2 days after the last medium change) and levels of secreted COMP were evaluated. Levels of COMP were estimated using COMP ELISA (Biovendor, Brno Czech Republic) according to the manufacturer’s instructions. The medium was not diluted during the assay. The absorbance was measured at 450 nm using the spectrophotometer SpectraMax i3 (Molecular Devices, San Jose, CA, USA).

### 2.6. Cartilage Explant Isolation and Co-Culture with MenSCs

For cartilage explants/MenSCs co-culture studies, MenSCs were detached, counted, and seeded into 6 well plates (100,000/well), while fragments of human OA articular cartilage were obtained from patients (*n* = 5) undergoing joint replacement surgery at Vilnius Santaros Hospital (bioethics committee permission No 158200-14-741) in a sterile container with PBS (Sigma Aldrich, St. Louis, MO, USA). The cartilage was washed with PBS containing 2% PS (Thermo Fisher Scientific, Waltham, MA, USA) and transferred to a Petri dish. The cartilage was cut into flat, round explants of 3 mm diameter and 3 mm height and each explant was weighted. The cartilage explants were washed and transferred into 6 inserts of 6 well plates (4 μm pore size) (120 mg of cartilage into one insert) for 24 h to a 37 °C incubator with 5% CO_2_. Cartilage explants were cultured in a medium which was composed of high glucose (4.5 g/L) DMEM medium and 2% PS. The next day, 4 inserts with cartilage explants were transferred to 6 well plates with MenSCs, leaving 2 inserts without interaction with cells and having two wells of MenSCs as control without co-cultures. Also, to half of the wells, IL-1β (10 ng/mL) (Thermo Fisher Scientific, Waltham, MA, USA) was added for stimulating inflammatory response. Cartilage explant co-cultures with MenSCs were cultures for 21 days, replacing the medium twice a week and collecting it after 3, 7, and 21 days for GAG analysis. Moreover, three cartilage explant samples were removed from co-cultures after 7 days for gene expression analysis (RT-qPCR).

### 2.7. ELISA for TGF-β1, Activin A, BMP-2 and IGF-1

TGF-β1, activin A, BMP-2, and IGF-1 protein production were evaluated during co-culture studies of MenSCs and BMMSCs, as well as MenSCs and cartilage explant co-cultures under inflammatory conditions with IL-1β (10 ng/mL). Supernatants were collected from the pellets incubated without growth factors during chondrogenic differentiation after 3, 7, and 21 days, and from explants after 21 days of incubation. TGF-β3-stimulated samples were not assessed in this study. Protein levels were detected using TGF-β1 (Biolegend), acitivin A (Abcam), BMP-2 (Abcam), and IGF-1 (Thermo Fisher Scientific) ELISA kits, according to manufacturer’s instructions. Protein levels secreted during co-cultures were normalized according to an upgrowing number of cells.

### 2.8. Glycosaminoglycan Analysis

The release of GAGs was assessed in the supernatants of cartilage explants incubated with and without MenSCs co-cultures and IL-1β for 3, 7, and 21 days. Measurements were performed using colorimetric 1,9-dimethyl-methylene blue dye (Bicolor Life Science assays, Carrickfergus, UK) dye, according to the manufacturer’s instructions. The standard used for GAG calibration was provided in the kit as a bovine tracheal chondroitin 4-sulfate. The absorbance was measured at 656 nm using the spectrophotometer, SpectraMax i3.

### 2.9. Histology and Immunohistochemistry

For the histochemical and immunohistochemical analysis of cartilage explants, the samples were fixed in 10% of neutral formalin and embedded into paraffin. The 4-micrometer sections were deparaffinized and stained with toluidine blue and safranin-O (pH 2.0) for 3 min. The safranin O stains GAGs and cell nuclei orange/red, while toluidine blue identifies cartilage ECM by staining it blue. The immunohistochemical staining with antibodies against collagen type II (Abcam, Cambridge, UK) was performed after the antigen retrieval with a citrate buffer pH 6.0 at +98 °C for 20 min and endogenous peroxidase blocking with 0.3% hydrogen peroxide for 15 min at room temperature (RT). The avidin–biotin complex (ABC) staining kit (Santa Cruz Biotechnology, Dallas, TX, USA) and 3.3-diaminobenzidine as a chromogen were used. Stained sections were evaluated and blindly scored independently by two histology experts.

### 2.10. RNA Extraction from the Cartilage Explant Samples after Co-Culturing with MenSCs

After 7 days of co-culturing cartilage explants with MenSCs, explants were collected, flash-frozen in liquid nitrogen, and stored at −70 °C. Frozen samples were homogenized using an ultrasonication system (Bandelin Sonopuls, Burlington, MA, USA) in lysis buffer (Qiagen, Venlo, The Netherlands) and RNA was extracted according to the manufacturer’s protocol. The RNA concentration and purity of all samples were measured with a SpectraMax i3 (Molecular Devices, San Jose, CA, USA).

### 2.11. RT-qPCR

RNA was reverse-transcribed with a Maxima cDNA synthesis kit, including dsDNase treatment (Thermo Fischer Scientific, Waltham, MA, USA). RT-qPCR reaction mixes were prepared with the Maxima Probe qPCR Master Mix (Thermo Fischer Scientific, Waltham, MA, USA) and TaqMan Gene expression Assays (RPS9—Hs02339424_g1, B2M—Hs00984230_m1, COL2A1—Hs01060345_m1, ACAN—Hs00153936_m1, SOX9—Hs00165814_m1, MMP13—Hs00233992_m1, MMP3—Hs00968305_m1, and run on the QuantStudio 1 Real-Time PCR System in technical triplicates starting with denaturation step at 95 °C for 10 min followed by 40 cycles at 95 °C for 15 s of denaturation and 60 s for annealing and extension. Relative levels of gene transcripts were calculated by subtracting the threshold cycle (Ct) of the normalizer (the geometric mean of the two house-keeping genes—RPS9 and B2M) from the Ct of the gene of interest, giving dCt values that were subsequently transformed to 2−dCt values and multiplied by 1000 to scale-up for better graphical representation.

### 2.12. Statistical Analysis

The Student’s *t*-test was used to calculate statistical significance. Data were considered to be statistically significant at *p* ≤ 0,05. PRISM8 software was used to perform statistical analysis. Not less than three patients’ cells and three repeats were measured.

## 3. Results

### 3.1. Stimulation of MenSCs and BMMSCs Chondrogenic Differentiation with TGF-β3, Activin A, BMP-2, and IGF-1

MenSCs and BMMSCs have been previously characterized according to stem cell markers (flow cytometry) and adipogenic and osteogenic differentiation capacity (Oil-Red/Alizarin) in our previous study [23]. We have demonstrated that both MenSCs and BMMSCs are positive (more than 95% of the total cell population) for classical MSC surface markers (CD44, CD73, CD90, CD105) and negative (less than 10% of the total cell population) for hematopoietic stem cell marker (CD14, CD34, CD36, CD45) expression. The proliferation rate of MenSC and osteogenic differentiation was superior as compared to BMMSC, however the adipogenic differentiation capacity of MenSCs was lower compared to BMMSCs.

In this experiment, chondrogenic differentiation of MenSCs and BMMSCs was stimulated in a pellet model using TGF-β3, activin A, BMP-2, IGF-1, and their combinations (TGF-β3 + activin A, TGF-β3 + IGF-1, and TGF-β3 + BMP-2) for 21 days. After differentiation, pellets were evaluated by size and prepared for gene expression analysis, as discussed in the methods section (see Section 2.2 and Section 2.3) (Figure 1). No differences in MenSCs pellet size were observed after stimulation with different growth factors, with all pellets being a similarly round shape (Figure 1A). However, differences in BMMSCs pellet size have been observed, where the largest pellets formed after stimulation with TGF-β3 only. Combinations of TGF-β3 + activin A, TGF-β3 + IGF-1, and TGF-β3 + BMP-2 also resulted in a slightly larger size as compared to single activin A, BMP-2, and IGF-1 stimulation in BMMSCs (Figure 1A). 

According to chondrogenic gene expression, the combination of TGF-β3 with Activin A, as well as TGF-β3 with IGF-1, resulted in significantly higher collagen type II gene expression in MenSCs compared to control and growth factors added alone. Also, TGF-β3 alone stimulated collagen type II gene expression in MenSCs, however, much weaker than its combination with activin A or IGF-1. A combination of TGF-β3 with activin A also significantly upregulated aggrecan gene expression in MenSCs compared to control and single growth factors. In BMMSCs, TGF-β3 significantly stimulated SOX9, aggrecan, and collagen type II. However, combinations of TGF-β3 + activin A, TGF-β3 + IGF-1, and TGF-β3 + BMP-2 revealed insignificantly higher amounts of aggrecan gene expression, but significantly upregulated collagen type II and SOX9 as compared to growth factors used alone (Figure 1B). Gene expression results supported pellet sizes in BMMSCs.

Taking together, this experiment demonstrates that the combination of TGF-β3 and activin A activate a MenSCs chondrogenic response that is stronger than other growth factors, while TGF-β3 is the major chondrogenesis stimulating factor for BMMSCs. BMP-2 and IGF-1 did not activate chondrogenic gene expression in both cell types if used alone. Even though the chondrogenic differentiation capacity of MenSCs was much weaker compared to BMMSCs, they possess an exclusive potential to differentiate into the chondrogenic lineage, which is different than in BMMSCs. This study is an extension of our previous study, where we demonstrated the potential of activin A to stimulate MenSCs chondrogenesis [23].

### 3.2. Chondrogenic Differentiation of BMMSCs Was More Pronounced in Co-Cultures with MenSCs According to COMP Secretion

Co-culture conditions were used for BMMSCs chondrogenic differentiation study, seeding MenSCs at the bottom of the wells and BMMSCs to 0.4 μm inserts, as discussed in the methods (see Section 2.4). COMP may be used as an indicator for chondrogenic differentiation in cells and its synthesis reveals the formation of cartilage ECM (17). COMP secretion after 21 days of differentiation was also significantly higher in co-culture conditions even without TGF-β3 added. The addition of TGF-β3 stimulated chondrogenesis even more in co-cultures (Figure 2).

Levels of secreted TGF-β1, activin A, BMP-2, and IGF-1 were analyzed in the medium after 3, 7, and 21 days of chondrogenic induction of BMMSCs in co-cultures with MenSCs. Both TGF-β1 and activin A levels were significantly higher in co-culture conditions as compared to single BMMSCs and single MenSCs cultures during 3 and 7 days, while BMP-2 secretion was significantly higher only after 21 days in co-cultures. IGF-1 did not reveal any significant differences between cells and co-cultures (Figure 3). These results indicate the importance of cell co-stimulation via growth factor secretion.

### 3.3. ECM Degradation in Cartilage Explants Was Decreased after Co-Culturing with MenSCs

Human articular cartilage explants were co-cultured with MenSCs to evaluate MenSCs paracrine factor effects on cartilage tissue ECM under inflammatory conditions with IL-1β. For these purposes, cartilage explants were cultivated in 0.4 μm inserts for 21 days, with/without MenSCs seeded in 6-well plate wells with/without IL-1β (as discussed in Section 2.6). As expected, the integrity of cartilage explant ECM was impaired after stimulation with IL-1β as compared to control samples. This was observed according to histological and immunohistochemical analysis as the staining intensity of safranin O, toluidine blue, and collagen type II antibody labelling decreased (Figure 4). On the other hand, explants cultivated in co-cultures with MenSCs demonstrated intense staining with safranin O and toluidine blue, as well as collagen type II antibody labelling. However, the effect of IL-1β was also visible in co-culture conditions with MenSCs with slightly reduced ECM staining, as compared to unstimulated with IL-1β co-cultures (Figure 4).

To analyze the amounts of released cartilage GAGs into the medium, the medium was collected after 3, 7, and 21 days of cartilage explant cultivation with MenSCs for GAG colorimetric assay. The results demonstrated similar tendencies to histological explant staining, with IL-1β increasing amounts of GAGs in the medium, while explant/MenSCs co-cultures significantly decreased the release of GAGs after 3, 7, and 21 days in culture (Figure 5). Interestingly, after 3 days of explant cultivation under co-culture with MenSCs and IL-1β conditions, a significantly lower release of GAGs was detected compared to explants stimulated with IL-1β only (Figure 5). These results present promising effects of MenSCs to further cartilage tissue regeneration studies.

After co-culturing cartilage explants with MenSCs under IL-1β stimulation for 21 days, the medium was collected and secretion of the same growth factors—TGF-β1, activin A, BMP-2, and IGF-1—were studied, as in BMMSCs/MenSCs co-cultures (Figure 6).

Cartilage explant chondrocytes demonstrated higher levels of secreted TGF-β1 as compared to co-cultures with MenSCs and MenSCs alone. IL-1β insignificantly decreased secretion of TGF-β1, however, additional stimulation with MenSCs increased the production of TGF-β1. Levels of activin A were significantly higher in co-cultures of explants with MenSCs, with and without IL-1β, while cartilage explants alone or under-stimulation with IL-1β did not reveal any significant differences. These results indicate the potential role of activin A in MenSCs co-activation with cartilage explant ECM and chondrocytes, which corresponds to BMMSCs/MenSCs co-activation in co-cultures. BMP-2 also showed tendencies of higher secretion in explant/MenSCs co-cultures with/without IL-1β, however, the results were insignificant. As for IGF-1, its levels were the highest in cartilage explants cultivated alone, while IL-1β, as well as co-cultures with MenSCs with/without IL-1β, have demonstrated a significant decrease of IGF-1. Moreover, explants in co-cultures with MenSCs with/without IL-1β showed significantly lower levels of IGF-1 than cartilage explants stimulated with IL-1β only (Figure 6).

After 7 days of cartilage explant co-culture with MenSCs and IL-1β, part of the samples were collected for gene expression analysis. It was decided to analyze the genes in the early stages of cultivation, while the effect might be most pronounced. Collagen type II, as well as matrix metalloproteinases 3 and 13 (MMP3 and MMP13) gene expression, are presented in Figure 7. All three genes revealed significant differences as a response to IL-1β with and without co-cultures with MenSCs, with collagen type II gene (COL2A1) expression being significantly downregulated, while MMP13 and MMP3 were significantly upregulated. MenSCs did not affect gene expression in chondrocytes, however, tendencies of increased collagen type II gene expression, as well as reduced MMP13 and MMP3 gene expression under stimulation with IL-1β, were observed. Data indicate that IL-1β negatively changes the chondrocyte gene expression profile, downregulating most abundant collagen type II synthesis and stimulating the release of degradation factors (MMP13 and MMP3), while MenSCs tends to lower its effect, which is a promising aspect for future studies.

## 4. Discussion

MSCs are non-hematopoietic cells with self-renewal abilities and are considered a promising approach for articular cartilage regeneration [24]. Different sources of MSCs, including adipose tissue, umbilical cord, placenta, dental pulp, and bone marrows, have been studied as cell therapies for repairing cartilage tissue in vitro and in vivo using different technological methods—scaffolds, hydrogels, and cell sheets [25,26]. BMMSCs are the most promising cell type for cartilage tissue regeneration due to their high potential to differentiate into chondrogenic lineage [27,28]. However, these cells are collected invasively and only small amounts of BMMSCs can be obtained from collected tissue. Therefore, other sources of MSCs for cartilage tissue repair are also in high demand and studies. 

MenSCs were first isolated in 2007 [29], and over the last 16 years, the therapeutic potential of MenSCs has been considered in multiple studies, such as neural, cardiac, liver, and lung diseases [15,30,31]. It was shown that MenSCs possess more advantageous properties compared to BMMSCs, as they are easy to harvest, differentiate into a number of tissue cells, have a high proliferative rate, and have low immunogenicity [32]. Also, MenSCs secrete large amounts of paracrine factors, including growth factors responsible for endometrium regeneration, which might be a potential co-stimulant for other tissue regeneration purposes [13,32]. Growth factors play an important role in regulating cellular functions, stimulating metabolic activity, and promoting stem cell differentiation. Even though MenSCs secrete a number of advantageous factors, their chondrogenic differentiation potential is still under question. A few studies reported that MenSCs are capable of differentiating into chondrogenic lineage according to cartilage ECM synthesis staining [33,34], however much more research should be implicated in order to verify their suitability for cartilage tissue repair.

In our previous study [23], we analyzed the chondrogenic differentiation capacity of MenSCs and BMMSCs by using two different growth factors—TGF-β3 and activin A—as an important growth factor in regulating women’s menstrual cycles [35,36,37]. Our results demonstrated different effects of activin A in two stem cell types, with higher stimulation in MenSCs. Therefore, in this study we included two additional growth factors—BMP-2 and IGF-1—to expand the growth factor spectrum and observe the role of BMP-2 and IGF-1 in stimulating MenSCs chondrogenesis. BMP-2 is a member of the TGF-β family and plays an important role in cartilage and bone tissue formation. It is known that BMP-2 induces chondrogenic and osteogenic differentiation in various types of stem cells via SOX9 and Runx2, however, it also induces endochondral ossification in MSCs [6,38]. The canonical BMP-2 pathway acts via activation of SMAD1/5/9, while TGF-βs and activins stimulate chondrogenesis via SMAD1/5/8 and SMAD2/3/4 [39,40,41]. BMP-2 can be used in cartilage tissue engineering but has to be properly regulated to retain cartilage phenotype. This is why the combination of BMP-2 and TGF-β3 was shown to be more effective and enhance the chondrogenic capacity of BMMSCs and adipose-derived MSCs, and blocking BMP-2 by noggin resulted in a less superior chondrogenic phenotype [42,43]. IGF-1 possess a different mechanism of action than BMPs and TGFs, as it was shown to act via type I tyrosine kinase receptor that activates the mitogen-activated protein kinase ½, extracellular signal-regulated kinase ½, mitogen-activated protein kinase, and the phosphatidylinositol-3-kinase-Akt pathways [44]. However, IGF-1 was shown to induce collagen type II, aggrecan, and SOX9 alone, and greater induction was observed in combinations with TGF-β1 in adipose-derived MSCs in vitro and in vivo [45], as well as induced chondrogenic differentiation, proliferation, and regulated apoptosis of mice BMMSCs [44].

In the current study, we were mostly focused on MenSCs paracrine factors and their effects on stimulating BMMSCs chondrogenic differentiation and protecting cartilage tissue explants from degradation. The initial step of the current research was to compare MenSCs and BMMSCs chondrogenic differentiation capacity by using four different growth factors, TGF-β3, activin A, BMP-2, and IGF-1, which are important for chondrogenesis. We have used these growth factors in combination with TGF-β3 by adding activin A, BMP-2, and IGF-1 only at the beginning of differentiation. This was made on purpose due to the importance of these growth factors in the early stages of chondrogenesis [8,46]. We used each growth factor separately to better understand their effects. Later, we evaluated the secretion of these growth factors in co-culture conditions of MenSCs with BMMSCs and cartilage explants. 

After chondrogenic differentiation of both cell types was stimulated using four different growth factors (TGF-β3, activin A, BMP-2 and IGF-1) and their combinations, we analyzed chondrogenic gene expression profile (SOX9, collagen type II, aggrecan) (RT-qPCR) in cell pellets and, similarly to our previous study, the chondrogenic differentiation capacity of MenSCs was much weaker compared to BMMSCs. However, the combination of TGF-β3 with activin A increased collagen type II and aggrecan gene expression, while in BMMSCs only TGF-β3 resulted in significant upregulation of chondrogenic genes (Figure 1). The addition of IGF-1 to differentiating BMMSCs or MenSCs did not significantly promote chondrogenic gene expression. As it has been reported in previous studies, IGF-1 has less effect on MSCs, or does only in combination with TGF-βs, as insulin is already present in chondrogenic medium with adding insulin-transferrin-selenium [44]. 

BMMSCs chondrogenesis was then stimulated in co-culture conditions with MenSCs. During this study, only TGF-β3 was used, and differentiation capacity was evaluated according to the secretion of COMP, TGF-β1, activin A, IGF-1, and BMP2 (ELISA). The results demonstrated that chondrogenic differentiation of BMMSCs was more pronounced in co-culture with MenSCs as compared to single BMMSCs cultures. COMP was used as the chondrogenic differentiation marker and showed a significant increase in its secretion in co-cultures, with and without TGF-β3 being added to the medium (Figure 2). Furthermore, the secretion of all growth factors was increased in cell co-cultures as compared to single MenSCs/BMMSCs populations (Figure 3). Co-culture conditions are common for MSCs and chondrocyte co-stimulation. Different types of MSCs were used for co-cultures together with chondrocytes (adipose tissue MSCs, fat pad adipose-derived MSCs and synovium MSCs) to stimulate cartilage formation [47,48]. Co-cultures of BMMSCs and chondrocytes already showed increased chondrogenesis by increased production of GAGs and increased Col II and ACAN gene transcription because of the continuous supply of growth factors, provided by BMMSCs [49,50]. According to the literature, MenSCs were never used in co-cultures with other cells for testing chondrogenic differentiation or cartilage explants. However, MenSCs encapsulated in fibrin glue were shown to repair osteochondral defects in rabbits [51], which is a promising result for further investigation.

Furthermore, we co-cultured human cartilage explants with MenSCs for 21 days under inflammatory conditions, stimulating with IL-1β, and evaluated cartilage explant ECM histologically, measuring GAG released into medium (colorimetric assay) and by collagen type II, as well as metalloproteinases 3 and 13 gene expression. According to histological and immunohistochemical analysis, cartilage explant ECM remained less damaged compared to control and samples with IL-1β (Figure 4). GAG analysis also showed a significant decrease of GAG release into the medium after co-culturing explants with MenSCs for 3, 7, and 21 days (Figure 5) as compared to explants cultured without MenSCs, suggesting a protective effect of MenSCs on cartilage ECM. Similar results were demonstrated with equine derived muscle MSCs cultivated in co-cultures with OA cartilage explants, where significantly lower amounts of GAGs were present in the medium of co-cultures compared to cartilage explants cultured alone [52]. The secretion of growth factors revealed significantly higher levels of activin A in co-culture medium with and without IL-1β and significantly decreased levels of IGF-1 in co-cultures (Figure 6). Also, BMP-2 and TGF-β1 showed tendencies of increased secretion in co-cultures of explants with MenSCs under stimulation of IL-1β. The development, growth, and maintenance of articular cartilage highly depend on the crosstalk of growth factors, such as TGF-βs, BMPs, IGFs, and activins, which properly regulate reparative functions of cartilage [53,54]. The role of IGF-1 in cartilage tissue development, ECM production, and chondrocyte metabolism has been reported in a number of studies [55]. IGF-1 can inhibit the production of IL-1β and metalloproteinase in chondrocytes [56], which is beneficial to cartilage tissue regeneration. Activin A is another important growth factor in cartilage development, and is an anti-catabolic molecule in articular cartilage [57]. The levels of IGF-1 were the highest in cartilage explants only, as compared to IL-1β stimulated samples and co-cultures with MenSCs. These results may suggest that IGF-1 is not involved in reparative MenSCs action, which is related to the activation of activin A signaling pathways. 

Collagen type II, MMP13, and MMP3 gene expression were analyzed in cartilage explants after co-culturing with MenSCs in order to evaluate MenSCs effects on chondrocyte metabolic gene expression. Even though the differences did not reach the level of statistical significance, MenSCs showed tendencies of upregulated COL2A1 expression and decreased MMPs under stimulation with IL-1β in co-cultures with explants (Figure 7). In our previous study, we co-cultivated human cartilage explants with adipose-derived MSCs and demonstrated a similar response of decreased MMP gene expression in chondrocytes after co-culturing with MSCs, suggesting beneficial paracrine factor action, promising developing novel therapeutic strategies based on MSC paracrine factors [58].

Thus, MenSCs may turn out to be a promising population of stem cells for their rich secretion of growth factors with the capacity to stimulate BMMSCs chondrogenic differentiation and prevent cartilage tissue from degradation. 

Study limitation: Even though our research provides important information regarding MenSCs paracrine effects on chondrogenic differentiation and cartilage tissue repair, it was focused mainly on TGF-β3, activin A, BMP-2, and IGF-1 growth factor stimulation and secretion. An additional study comparing other growth factors, signaling molecules, and their molecular pathways that participate in MSC and chondrocyte cross-talk would strengthen the current study. Also, a broadened understanding of how MenSCs EVs affect BMMSCs and cartilage explants could indicate their potential for future non-cellular therapy.

## 5. Conclusions

The results suggest the stimulatory effects of MenSCs on chondrogenic differentiation of BMMSCs, as it was more pronounced in co-culture with MenSCs. The secretion of TGF-β1, activin A, and COMP was higher in cell co-cultures as compared to single populations, which may stimulate cartilage-related ECM production in BMMSCs. Human cartilage explants co-cultured with MenSCs resulted in increased secretion of activin A, while it downregulated IGF-1 and reduced cartilage ECM degradation under inflammatory conditions. 

MenSCs may turn out to be a promising population of easily accessible stem cells for the development of cell-based therapies with the capacity to stimulate BMMSCs chondrogenic differentiation and prevent cartilage tissue from degradation.

## Figures and Tables

**Figure 1 bioengineering-10-01001-f001:**
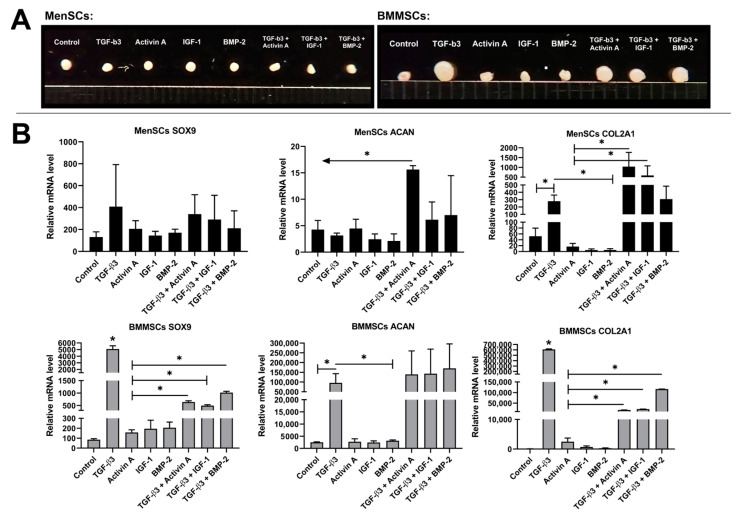
Chondrogenic differentiation of MenSCs and BMMSCs for 21 days in chondrogenic medium with TGF-β3 (10 ng/mL), activin A (50 ng/mL), BMP-2 (50 ng/mL), and IGF-1 (100 ng/mL), and their combinations TGF-β3 + activin A, TGF-β3 + IGF-1, and TGF-β3 + BMP-2, where activin A, IGF-1, and BMP-2 were added only during the first two days. (**A**)—Morphological analysis of BMMSCs and MenSCs pellet size, macroscopic view. (**B**)—chondrogenic gene (SOX9, aggrecan (ACAN), collagen type II (COL2A1)) expression in MenSCs (*n* = 4), and BMMSCs (*n* = 3) pellets (RT-qPCR). Control—cells differentiated in the chondrogenic medium without growth factors. Relative mRNA level presented after normalization to two housekeeping genes (B2M and RPS9) and were expressed as 2-ΔCt*1000. Data are presented as mean ± SD. * and horizontal bars represent *p* ≤ 0.05, while arrows represent the significant difference of *p* ≤ 0.05 in the presented direction.

**Figure 2 bioengineering-10-01001-f002:**
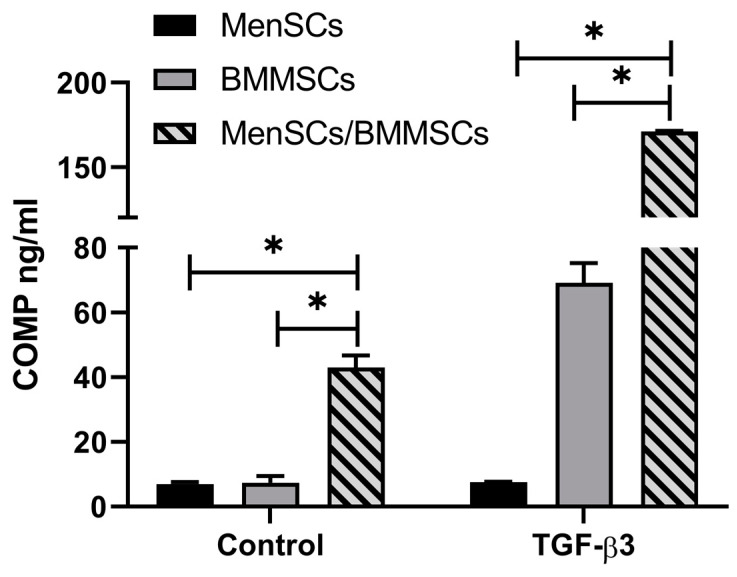
Levels of secreted COMP protein in the chondrogenic medium after 21 days of BMMSCs (*n* = 3) differentiation under co-culture conditions with MenSCs (*n* = 4) (using 0.4 μm inserts), without (Control) and with TGF-β3 (ELISA). Data are presented as mean ± SD. * Horizontal bars represent *p* ≤ 0.05.

**Figure 3 bioengineering-10-01001-f003:**
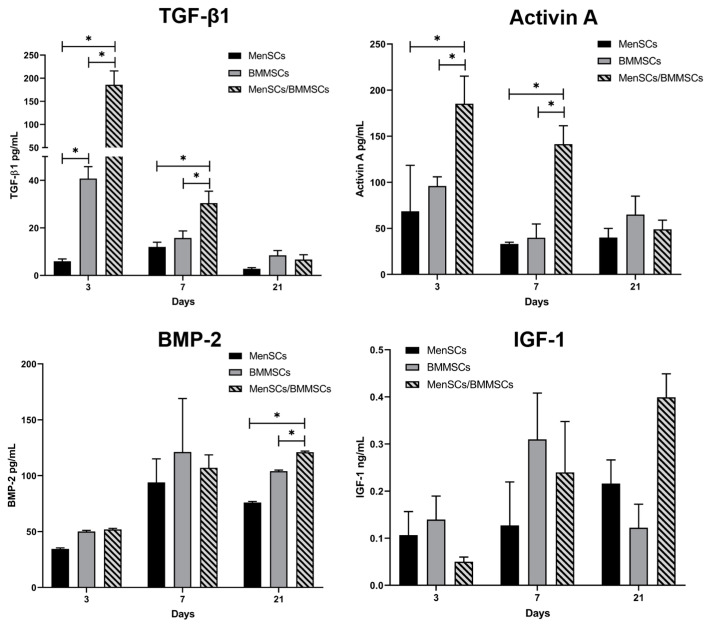
Levels of secreted TGF-β1, activin A, BMP-2, and IGF-1 after 3, 7, and 21 days of chondrogenic differentiation of BMMSCs (*n* = 3) in co-cultures with MenSCs (*n* = 4), using 0.4 μm inserts in chondrogenic medium without growth factors (ELISA). Data are presented as mean ± SD. * Horizontal bars represent *p* ≤ 0.05.

**Figure 4 bioengineering-10-01001-f004:**
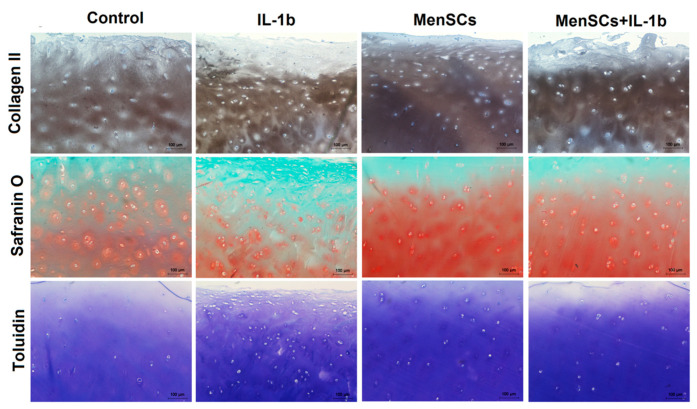
Histological and immunohistochemical analysis of cartilage explants in co-cultures with MenSCs, using 0.4 μm inserts in cartilage medium after stimulation with IL-1β (10 ng/mL) for 21 days. Staining with safranin O, toluidine blue, and labelling with collagen type II antibodies. X40. Control—explants without co-culturing with MenSCs and without stimulation with IL-1β.

**Figure 5 bioengineering-10-01001-f005:**
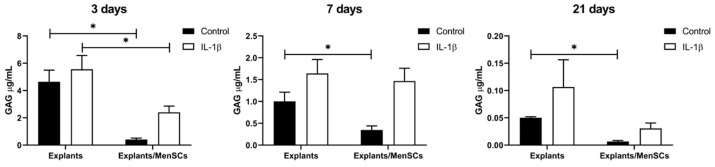
Glycosaminoglycan (GAG) release in cartilage explants after co-culturing with MenSCs, using 0.4 μm inserts in cartilage medium and stimulation with IL-1β (10 ng/mL) for 3, 7, and 21 days (colorimetric assay). Data are presented as mean ± SD. * Horizontal bars represent *p* ≤ 0.05.

**Figure 6 bioengineering-10-01001-f006:**
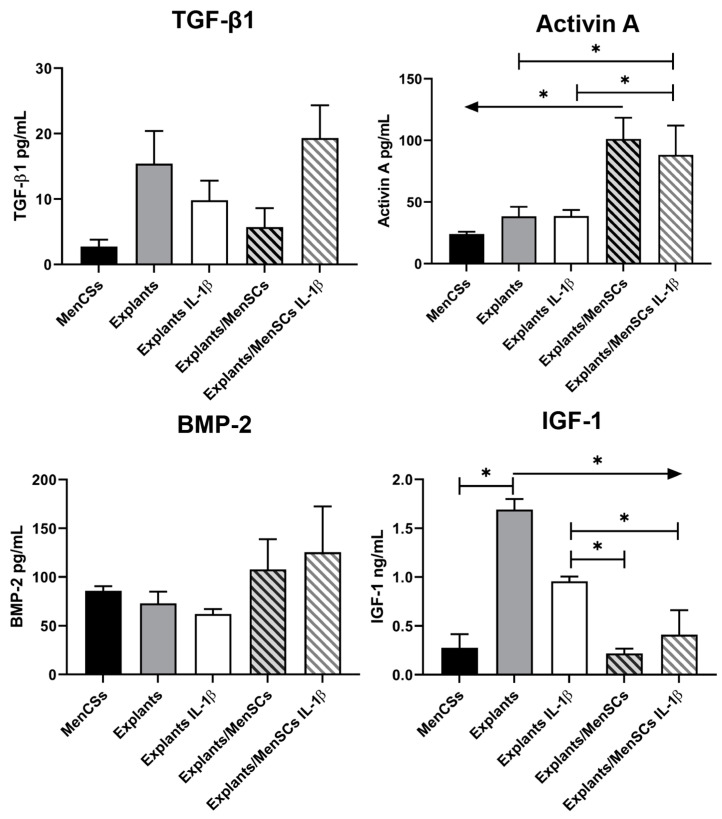
Levels of secreted TGF-β3, activin A, BMP-2, and IGF-1 after 21 days of cartilage explant (*n* = 5) cultivation in co-cultures with MenSCs (*n* = 4), using 0.4 μm inserts in cartilage medium after stimulation with IL-1β (10 ng/mL) (ELISA). Data are normalized to MenSCs amount and presented as mean ± SD. * Horizontal bars represent *p* ≤ 0.05, while arrows represent a significant difference of *p* ≤ 0.05 in a specific direction, while arrows represent a significant difference of *p* ≤ 0.05 in a specific direction.

**Figure 7 bioengineering-10-01001-f007:**
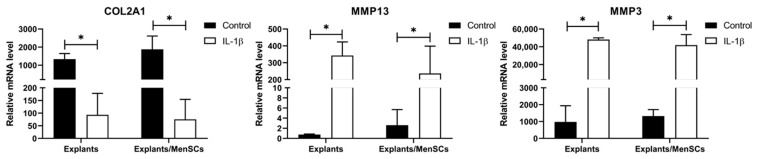
Expression of collagen type II (COL2A1), metalloproteinase 13 (MMP13) and metalloproteinase 3 (MMP3) genes in cartilage explant chondrocytes (*n* = 5), after co-culturing with MenSCs (*n* = 4), using 0.4 μm inserts in cartilage medium and stimulation with IL-1β (10 ng/mL) for 7 days (RT-qPCR). Relative mRNA level presented after normalization to two housekeeping genes (B2M and RPS9) and expressed as 2-ΔCt*1000. Data are presented as mean ± SD. * Horizontal bars represent *p* ≤ 0.05.

## Data Availability

The data supporting these findings can be found at State Research Institute Centre for Innovative Medicine, Department of Regenerative Medicine.

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
