# Peer review of "Menstrual Blood-Derived Stem Cell Paracrine Factors Possess Stimulatory Effects on Chondrogenesis In Vitro and Diminish the Degradation of Articular Cartilage during Osteoarthritis"

_bioengineering, 2023, doi:10.3390/bioengineering10091001_

Round 1

Reviewer 1 Report

The manuscript by Uzieliene, et al. delves into the potential of menstrual blood-derived mesenchymal stem cells (MenSCs) in stimulating bone marrow mesenchymal stem cells (BMMSCs) towards chondrogenic differentiation, as well as investigating their role in protecting cartilage from degradation. The researchers induced chondrogenic differentiation in BMMSCs using four growth factors, and MenSCs were co-cultured with BMMSCs and cartilage explants. The study's findings highlight that MenSCs play a pivotal role in stimulating the chondrogenic response in BMMSCs by secreting activin A and TGF-β3. Furthermore, they show that MenSCs may have protective effects on cartilage tissue extracellular matrix (ECM) by reducing the release of GAGs, potentially through modulation of activin A-related molecular pathways. The results demonstrate that MenSCs' paracrine factors hold great promise as a therapeutical approach for cartilage tissue protection and repair. This research opens up new avenues for cartilage tissue engineering using MenSCs however, further studies are necessary to validate and understand the underlying mechanisms fully. Overall, this study offers an exciting glimpse into the potential of MenSCs in the field of cartilage tissue engineering and regenerative medicine.

Here are few comments and feedback on the Materials and Methods:
1. Isolation and Culture of MenSCs and BMMSCs: The description of MenSCs and BMMSCs isolation and culture is concise and clear, making it easy for readers to grasp the process. However, additional information on the culture conditions (e.g., cell density, medium volume, frequency of medium changes) for both cell types will be helpful.
2. Chondrogenic Differentiation in Pellets with Different Growth Factors: The methodology for this section is well-described. It would be beneficial to clarify the rationale for choosing specific growth factor combinations (e.g., TGF-β3 + activin A, TGF-β3 + BMP-2, TGF-β3 + IGF-1) and why TGF-β3 alone was not assessed for protein production.
3. Glycosaminoglycan Analysis: It would be valuable to mention the specific GAG standard used for calibration.

Given IGF-1 plays a crucial role in promoting chondrogenic differentiation by stimulating cell proliferation, regulating ECM synthesis, and modulating various signaling pathways. It is important to discuss how changes in IGF-1 levels plays a role in chondrogenic differentiation and how the source of MSCs, and the chondrogenic induction can affect this process.

Additionally, it would be beneficial for the authors to consider the following comments:

- Integration with previous studies: While the authors cite previous research regarding MenSCs and their advantages over BMMSCs, they could further strengthen the discussion by comparing their results with existing literature. Discussing similarities or differences with other studies will provide context and contribute to the current understanding of MenSCs' potential for cartilage tissue repair.

- Limitations and future directions: The authors should acknowledge any limitations of their study and suggest potential future research directions. Addressing the limitations will enhance the credibility of the findings and encourage further investigation in the field.

- Mechanistic insights: The authors could enrich the discussion by delving into the potential underlying mechanisms through which MenSCs stimulate BMMSCs' chondrogenic differentiation and protect cartilage ECM. Speculating on the signaling pathways and interactions between cells and growth factors could add depth to the interpretation of results.

There are a few instances of incorrect grammar and punctuation errors throughout the manuscript. For instance, there are missing commas in some sentences, and subject-verb agreement issues in others. A thorough proofreading would help to correct these minor errors.

In case pf sentence structure, the text contains some long and complex sentences, making it slightly challenging to follow at times. Breaking down these sentences into shorter, more concise statements would enhance readability.

Author Response

Answer: We thank the reviewer for his highlights and valuable suggestions. We have tried to address every comment and correct the manuscript accordingly. Please follow our answers and check the revised manuscript for more clarity. Added information is marked in blue.

Here are few comments and feedback on the Materials and Methods:
1. Isolation and Culture of MenSCs and BMMSCs: The description of MenSCs and BMMSCs isolation and culture is concise and clear, making it easy for readers to grasp the process. However, additional information on the culture conditions (e.g., cell density, medium volume, frequency of medium changes) for both cell types will be helpful.

Answer: Thank you, we have included more details in the methods section to make it clearer for readers.

  1. Chondrogenic Differentiation in Pellets with Different Growth Factors: The methodology for this section is well-described. It would be beneficial to clarify the rationale for choosing specific growth factor combinations (e.g., TGF-β3 + activin A, TGF-β3 + BMP-2, TGF-β3 + IGF-1) and why TGF-β3 alone was not assessed for protein production.

Answer: Thank you for this comment. We have added information about growth factors and their combinations in the discussion. The rationale for choosing such growth factor combination by adding different growth factor separately was to better understand each growth factor effect. Regarding TGF-β3: it is an isoform of TGF-β1 and they both act stimulating chondrogenesis in MSCs, with TGF-β3 being exclusively effective. However, as TGF-β1 is abundantly secreted by chondrocytes, we decided to check the TGF-β1 secretion by MSC during chondrogenesis in co-cultures and cartilage explants.

  1. Glycosaminoglycan Analysis: It would be valuable to mention the specific GAG standard used for calibration.

Answer: The standard used within this kit was a bovine tracheal chondroitin 4-sulfate. We have added this information in the methods section (2.1.).

Given IGF-1 plays a crucial role in promoting chondrogenic differentiation by stimulating cell proliferation, regulating ECM synthesis, and modulating various signaling pathways. It is important to discuss how changes in IGF-1 levels plays a role in chondrogenic differentiation and how the source of MSCs, and the chondrogenic induction can affect this process. 

Answer: Thank you, we have added more information regarding IGF-1 and its mechanism of action in the discussion:

Increase of IGF-1 is associated with anabolic processes, including proliferation, stimulation of ECM synthesis and etc. During chondrogenic differentiation of MSCs, IGF-1 regulates cell apoptosis, promotes proliferation, and induces chondrogenic phenotype. According to our results, addition of IGF-1 to differentiating BMMSCs or MenSCs did not significantly promote chondrogenic geneexpression. This might be the case that other growth factors should be added to observe significant increase, or significant amount of IGF-1 is already synthesized by cells so adding more do not result in stimulation of chondrogenic genes. It was shown before that IGF-1 has a stronger effect when added together with TGF (Longobardi 2006), as it acts via different pathway (not canonical SMAD activation like TGF/BMP/activins do). However, IGF-1 was shown to induce collagen type II, aggrecan, and SOX9 alone and greater induction was observed in combinations with TGF-β1 in adipose-derived MSCs in vitro and in vivo.

Additionally, it would be beneficial for the authors to consider the following comments:

- Integration with previous studies: While the authors cite previous research regarding MenSCs and their advantages over BMMSCs, they could further strengthen the discussion by comparing their results with existing literature. Discussing similarities or differences with other studies will provide context and contribute to the current understanding of MenSCs' potential for cartilage tissue repair.

Answer: Thank you for pointing this out. We have expanded the discussion comparing our results with other studies. Regarding MenSCs, they were less studied for cartilage tissue regeneration/chondrogenic differentiation, as compared to adipose or bone marrow MSCs. According to the current literature, MenSCs were never used in co-cultures with chondrocytes or cartilage explants before. We have added information about application of other sources of MSCs co-cultures with chondrocytes and included a study where MenSCs were applied to osteochondral defects in rabbits.

- Limitations and future directions: The authors should acknowledge any limitations of their study and suggest potential future research directions. Addressing the limitations will enhance the credibility of the findings and encourage further investigation in the field.

Answer: Thank you, we have added limitations of our study in the discussion part.

- Mechanistic insights: The authors could enrich the discussion by delving into the potential underlying mechanisms through which MenSCs stimulate BMMSCs' chondrogenic differentiation and protect cartilage ECM. Speculating on the signaling pathways and interactions between cells and growth factors could add depth to the interpretation of results.

Answer: Thank you, we have improved the discussion part. In the current study, we conclude that MenSCs stimulate chondrogenic response in BMMSCs by secreting activin A and TGF-β3, and may have protective effects on cartilage tissue ECM by decreasing release of GAGs, most likely through modulation of activin A related molecular pathway.

Comments on the Quality of English Language

There are a few instances of incorrect grammar and punctuation errors throughout the manuscript. For instance, there are missing commas in some sentences, and subject-verb agreement issues in others. A thorough proofreading would help to correct these minor errors.

In case pf sentence structure, the text contains some long and complex sentences, making it slightly challenging to follow at times. Breaking down these sentences into shorter, more concise statements would enhance readability.

Answer: Thank you for pointing this out. We have corrected minor mistakes and shortened long sentences in the manuscript.

Reviewer 2 Report

This is a very interesting study in which menstrual blood-derived MSCs (MenSCs) were examined for their ability to enhance chondrogenesis in co-culture with BMMSCs and cartilage explants. The authors demonstrate that indeed, MenSCs increase cartilage marker expression in stem cell pellet-based and cartilage explant-based culture models. This study demonstrates the potential for MenSCs in cartilage tissue engineering and regenerative medicine.

I think the manuscript could be strengthened by more thorough description of the characteristics of MenSCs - i.e. cell surface markers, growth kinetics, morphology, yield, etc. 

It may also be helpful to speculate as to why the addition of MenSCs produces such profound pro-chondrogenic/anti-inflammatory effects. Could this be attributed to the fact that these stem cells resident in the female reproductive system are routinely exposed to highly inflammatory microenvironments during menstruation? Is it known if MenSCs secrete estrogen or other hormones?

There are quite a few grammatical errors as well as instances in which colloquial English is used instead of formal. For example, on page 11, "Couple of studies reported.."

Author Response

We thank the reviewer for his/her valuable comments and suggestions. As we state in the manuscript, we have used MenSCs that we already characterized before by surface marker expression, proliferation/growth rate, morphology, etc. We have added the link to our previously published manuscript with all these characteristics into the result section. Added information is marked in blue.

It may also be helpful to speculate as to why the addition of MenSCs produces such profound pro-chondrogenic/anti-inflammatory effects. Could this be attributed to the fact that these stem cells resident in the female reproductive system are routinely exposed to highly inflammatory microenvironments during menstruation? Is it known if MenSCs secrete estrogen or other hormones?

Answer: Thank you for this comment. We absolutely agree on the fact, that MenSCs are being exposed to highly inflammatory microenvironment and therefore, they secrete a number of growth and anti-inflammatory factors to control the process.

As elucidated in the manuscript, MenSCs exhibit the capacity to secrete a diverse array of growth factors, which hold the potential to intricately influence chondrogenesis both directly and indirectly. One such pivotal factor, activin A, could be target of paracrine loop within BMMSCs and MenSCs. In our experiments the level of activin A was higher in MenSCs/BMMSCs co-cultures than in MenSC and BMMSCs cultures combined, thus indicating the importance of cellular co-activation and co-stimulation.

There might be a biological explanation for anti-inflammatory effects and high levels of growth factors in MenSCs. Menstrual cycle ushers in a phase of heightened inflammation in the female reproductive system, necessitating the restoration of uterine lining post-cycle. Within this context, the swift and robust proliferation, and anti-inflammatory properties of MenSCs become essential for rapid uterine tissue regeneration. As for female hormones they are produced in ovaries, meanwhile MenSCs react to changing levels of progesterone and estradiol during menstrual cycle.

Also, the uterus and its cells, including MenSCs, doesn’t synthesize and secrete any hormones, as they are produced by ovarian corpus luteum cyst and control women reproductive system.

Comments on the Quality of English Language

There are quite a few grammatical errors as well as instances in which colloquial English is used instead of formal. For example, on page 11, "Couple of studies reported.."

Answer: Thank you. We have corrected the errors, typos and informal language throughout the manuscript.

Reviewer 3 Report

Well written manuscript with no significant changes needed. Interesting concept of using MenSC for anabolic and antiinflammatory properties. 

Author Response

We thank the reviewer for his/her compliments regarding our manuscript. We hope it will broaden the field of using MenSCs for cartilage tissue regeneration.

Reviewer 4 Report

The authors Uzieliene et al. present an interesting article about Menstrual Blood-Derived Stem Cell Paracrine Factors and their use as stimulatory effect to Chondrogenesis in vitro. 

Chondrogenic differentiation of BMMSCs was stimulated in co-cultures with MenSCs and cartilage explants co-cultured with MenSCs for 21 days. Chondrogenic capacity of BMMSCs was analysed by secretion of four growth factors and cartilage oligomeric matrix protein, as well as release and synthesis of cartilage extracellular matrix proteins and gene expression in cartilage explants.

Although the study is interestingly structured and well presented, some questions remain for further clarification. 

1. Please describe the MenSC in more detail. A typing by FACS for the exact description of the cells contained in the MenSC is missing. If this has already been done in preliminary work, a better description in the introduction is sufficient. How many % of e.g. MSC, EPC etc. do the MenSC contain?

2. Mixing different cell types from different donors is considered critical in this experiment. Mixing stem cells from one donor with MenSC from other donors will always lead to a reaction. Please describe in detail to what extent the results are due to the actual cell reaction. Can an immunological reaction be excluded. If so, how can you prove it. 

Otherwise, the effects shown could also be due to other reasons and the result should be doubted. 

minor spellcheck

Author Response

  1. Please describe the MenSC in more detail. A typing by FACS for the exact description of the cells contained in the MenSC is missing. If this has already been done in preliminary work, a better description in the introduction is sufficient. How many % of e.g. MSC, EPC etc. do the MenSC contain?

Answer: We thank the reviewer for his/her valuable comments and suggestions. As we state in the manuscript, we have used MenSCs that we already characterized before by surface marker expression, proliferation/growth rate, morphology, etc. We have added the link to our previously published manuscript with all these characteristics into the result section and also added the percentage of MSC markers MenSCs possess. Added information is marked in blue.

  1. Mixing different cell types from different donors is considered critical in this experiment. Mixing stem cells from one donor with MenSC from other donors will always lead to a reaction. Please describe in detail to what extent the results are due to the actual cell reaction. Can an immunological reaction be excluded. If so, how can you prove it. 

Answer: Thank you for this comment. We need to note that there was no direct mixing of cell types during the study.

In this investigation, Menstrual-Derived Stem Cells (MenSCs) were co-cultured with either Bone Marrow Mesenchymal Stem Cells (BMMSCs) or cartilage explants using an insert method. This approach ensured that there was no direct contact between MenSCs and either BMMSCs or cartilage explants, mitigating the potential for any immunological reactions between biological materials from different donors. Furthermore, it is important to note that significant immunological reactions to donor Mesenchymal Stem Cells (MSCs) are typically observed in vivo or when culturing MSCs with immune cells such as dendritic cells or T cells in vitro. Since both MenSCs and BMMSCs are not immune cells, the likelihood of an immunological reaction occurring between the distinct biological materials of different donors is minimal. We have added this information into the discussion section together with links to papers. Also, MenSCs are negative for immune activation marker MHC class II antigen (HLA-DR) (Mou et al 2013, Chet et al 2019, Sanchez-Mata et al, 2021), so no they do not cause immunological reaction and they are suitable for allogenic application.

https://www.ncbi.nlm.nih.gov/pmc/articles/PMC8178850/

https://onlinelibrary.wiley.com/doi/10.1111/j.1365-2567.2008.02891.x

Round 2

Reviewer 4 Report

all questions were adequately answered 

there are no concerns regarding an acception